# Computational modeling demonstrates that glioblastoma cells can survive spatial environmental challenges through exploratory adaptation

Orieta Celiku [1], Mark R. Gilbert[1] & Orit Lavi[2]*

Glioblastoma (GBM) is an aggressive type of brain cancer with remarkable cell migration and adaptation capabilities. Exploratory adaptation—utilization of random changes in gene regulation for adaptive benefits—was recently proposed as the process enabling organisms to survive unforeseen conditions. We investigate whether exploratory adaption explains how GBM cells from different anatomic regions of the tumor cope with micro-environmental pressures. We introduce new notions of phenotype and phenotype distance, and determine probable spatial-phenotypic trajectories based on patient data. While some cell phenotypes are inherently plastic, others are intrinsically rigid with respect to phenotypic transitions. We demonstrate that stochastic exploration of the regulatory network structure confers benefits through enhanced adaptive capacity in new environments. Interestingly, even with exploratory capacity, phenotypic paths are constrained to pass through specific, spatial-phenotypic ranges. This work has important implications for understanding how such adaptation contributes to the recurrence dynamics of GBM and other solid tumors.

[1] Neuro-Oncology Branch, CCR, NCI, NIH, Bethesda, MD, USA. [2] Integrative Cancer Dynamics Unit, Laboratory of Cell Biology, CCR, NCI, NIH, Bethesda, MD, USA. *email: orit.lavi@nih.gov

Glioblastoma (GBM) is the most lethal adult primary brain cancer and remains incurable despite decades of research[1,2]. Despite the use of multi-modality therapeutic approaches, most GBM patients experience recurrence within 6–9 months of primary treatment; over 80% of the first recurrences occur at the original tumor site[3]. Recurrence requires three main steps: (1) existence of remnant tumor cells capable of surviving outside of the resected tumor area, 2) migration to distant locations or adaptation to new environments, and (3) formation of a new tumor by transitioning back to a phenotype with similar features to the primary tumor (such as increased growth rate and reduced motility). The ability of GBM cells to quickly undergo such transformation in the complex brain environment implies that these cells can adapt to conditions and stresses that they have not been pre-programed to handle; it is, therefore, reasonable to hypothesize that GBM cells could follow a collective process of adaptation regardless of their initial molecular state or physical location. An understanding of the fundamental processes of such biological adaptation and disruption of any of the involved steps may lead to a robust treatment approach that reduces the likelihood of patient relapse. In this work, we study these fundamental components of recurrence by studying the critical processes of phenotype trajectory, adaptation, phenotype transition, and reversal from one phenotype to another.

Two recent research developments encouraged us to consider the question of adaptation of spreading GBM cells. First, the Braun and Brenner groups[4,5] proposed an intriguing theory of exploratory adaption, which addresses the fundamental question of how organisms deal with unforeseen environments as a generic process. Braun et al.[6–9] subjected yeast and fruit flies to a series of novel challenges and observed a wide spectrum of adaptive responses, demonstrating that there is no pre-evolved mechanism to handling novel challenges; by comparison, known challenges were met with more uniform responses. They concluded that surviving novel challenges requires exploratory changes in gene regulation of individual organisms[5]. Brenner et al.[4] recently developed a theoretical model of exploratory adaptation that prescribes how small random perturbations to the gene-regulatory network could be propagated to changes in the cellular phenotype, and demonstrated the feasibility of this process for networks with certain topological characteristics. Second, the Ivy Glioblastoma Atlas Project (Ivy GAP) has recently constructed a transcriptional atlas of GBM that aligns tumor anatomical regions with histopathologic features[10].

We sought to understand how and when GBM cells use their intrinsic versus exploratory capacity to adapt to their environment, and how these processes can explain the fundamental dynamics of recurrence. We extended the exploratory adaptation model, including with notions of gradual adaptation. We studied exploratory adaptation in GBM using patient-derived transcriptomic profiles of spatially separated anatomical tumor structures from Ivy GAP. We introduced new notions of phenotype based on functional pathway activity (which reflects the degree of coordinated up or downregulation of the member genes' expression), and defined a measure of phenotype differences which we call phenotype distance. We identified three spatial trajectories that dominate the phenotypic diversity of the GBM locations. We investigated whether stochastic changes to the regulatory expression network could explain the cells' ability to adapt from the phenotype of one location to that of another along the identified trajectories. For example, we examined how cells from the tumor core (CT) can adapt to resemble those of the leading edge (LE) (Fig. 1a). We developed an optimization approach that models how exploratory cells approaching a target phenotype adapted to its environment may reduce their exploration and thus converge to that phenotype. We simulated

the GBM cellular responses to familiar and new environmental stimuli and showed, for example, that several possible continuous phenotype trajectories could be used by cells to spread from CT to LE. We estimated the distributions of the pathway activity over time, as well as a coordinated global measure of the cell's phenotype changes. This enabled us to observe instances of convergence: for example, we observed pathways whose activity distributions transitioned from multimodal to unimodal distributions overlapping with target distributions. We next investigated whether the process of adaptation is reversible (for example, whether LE cells can revert to resembling those of CT), and whether phenotypically-distinct spatially intermediate states (such as infiltrating tumor (IT) cells) are necessary during this transition. Finally, we propose hindering the adaptation process through targeting of intermediate phenotypes as a treatment roadmap.

## Results

**Framing the problem of exploratory adaptation in GBM cells.** We framed this research effort by addressing the following questions. Can GBM cells follow the exploratory adaptation process? What are the circumstances under which such exploration might happen? To what extent can phenotypic diversity of distinct tumoral regions be the result of such adaptation process? To address these questions, we introduced our definitions of phenotype, phenotype distance, and a dynamical model that simulates the changes a GBM cell may undergo to adapt from an initial phenotype to target phenotypes. Using this framework and patient-derived GBM data we carried out the following steps. First, we estimated a set of genes relevant to GBM, which were used to calculate location-specific phenotypes, and to estimate the distances between the phenotypes. We constructed a phenotype network with spatial locations as nodes and edges weighted by the estimated phenotype distances (Fig. 1a–c). Second, we inferred a set of important spatial phenotypic trajectories based on the shortest phenotype distance between the nodes of the network (Fig. 1d–i). Third, we developed a dynamic model to simulate the process of exploratory adaptation in GBM along the spatial trajectories (Fig. 2). Fourth, we used this model to estimate the intrinsic ability of each location to undergo spontaneous phenotypic changes over time. This serves as our control or expected behavior when no novel environment challenges are present (Fig. 3a). Fifth, we simulated how exploratory adaptation, through its effect on the regulatory network, impacts location phenotypes and their convergence to new phenotypes (Fig. 3b). Sixth, we explored whether phenotypic changes achieved through exploratory adaption are reversible (Fig. 4a). Seventh, we estimated possible intermediate phenotypes that such reversal might take (Fig. 4b–d). Eighth, we proposed a treatment roadmap that could mitigate the acquired plasticity (see Supplementary Data 2). Finally, we assessed the location-based phenotypes in terms of their immune signatures.

**Initial data exploration.** Directly addressing questions related to spatial adaptation processes would require time-series molecular profiles of GBM, but to our knowledge no such data are publicly available. Instead, we propose using data sampled from distinct regions within the same tumor to represent different snapshots of adaptation. The Ivy GAP[10] dataset contains transcriptomic profiles of distinct anatomic regions within the same patient tumor samples and is thus well suited to our task. First, we extracted a focused list of GBM-related genes by selecting the genes differentially expressed between low-grade gliomas (LGG) and GBM and high expression variability in the combined glioma cohort (selecting the top 10 percentile of variation) profiled by The

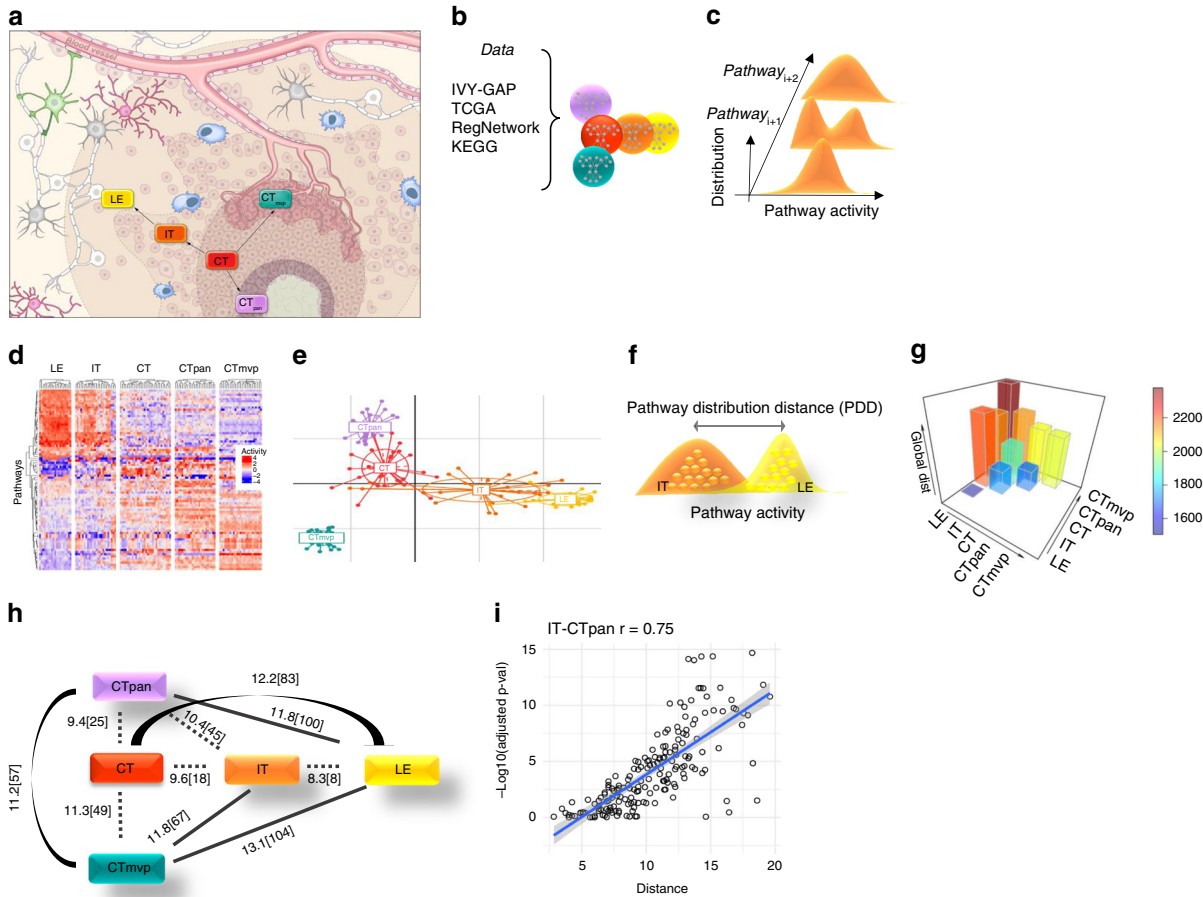

**Fig. 1 GBM spatial phenotypic trajectories.** Initial analysis of the static data was conducted to assess spatial phenotypic trajectories. Pathways and global distances were calculated for every pair of locations based on population distributions. **a** Illustration of sampled locations from Ivy GAP datasets, including gene expression data from 41 patients taken from the following anatomic regions: cellular tumor (CT), leading edge (LE), infiltrating tumor (IT), pseudopalisading region around necrosis (CTpan), and microvascular proliferation (CTmvp). Arrows demonstrate the potential spatial tumor spread patterns that we focused on throughout this study. **b** For each sample, a molecular network was constructed based on our GBM focused gene list, relevant TFs, gene expression, and pathway activities. **c** A sample's phenotype is defined to be its vector of pathway activities. For each location, a pathway activity distribution was created based on the pathway activities of all samples from that location. The location's phenotype was thus defined as a vector of pathway activities distributions. **d–g** Differences between phenotypes were estimated in several ways: **d** clustering groups of samples based on their pathway activity patterns (see Supplementary Fig. 1 for more details), **e** data reduction of pathway activities using BGA. The plot visualizes the spatial trajectories using patterns of pathway activities of all samples, **f** pathway distributions distances were calculated between every pair of locations. **g** integrating all those pathway distributions distances between every pair of locations gives a reduced value of global distances, plotted as histogram. **h**, **i** Resulting phenotype distances and phenotype spatial trajectories. This complex information can be depicted as a network, where nodes are location's phenotypes, and edges are distances between phenotypes. The edges' weights can be estimated in three ways: first (section H), mean of the pathways distribution distances (PD), given as the first value. Second (section I), number of differential pathway activities (DA), given as the second value in brackets, and third, global distances, given in Supplementary Data 1. Based on these three values, shortest paths are as follow: (CT, CTpan), (CT,IT), (CT, CTmvp), (CTpan,IT), and (IT, LE), are marked with dashed lines. All other larger distances are marked with solid lines. PD and DA are strongly correlated across all locations (see Supplementary Data 1 and Supplementary Fig. 1 for a complete list of p-values and correlation coefficients).

Cancer Genome Atlas (TCGA)[11,12]. The list was expanded to include any transcription factors (TFs) that regulate these GBM-focused genes. This list contains ~1/3 of the genes that are sufficiently expressed in the Ivy-GAP cohort. We assessed the overlap of our gene list with other glioma related sets by selecting the (25) glioma related gene sets curated as part of the Molecular Signatures Database (MSigDB)[13]. For each of these MSigDB gene sets we examined the percentage of expressed genes that overlap with our selected gene list. The majority of the glioma relevant gene sets are well represented in our selected list (see Supplementary Data 1).

Using TF-gene regulatory information from RegNetwork[14] we constructed a network with 4121 nodes (our focused genes) and 78,686 directed edges (average degree of 19.09). We then used the Ivy GAP dataset to assess the exploratory adaptation hypothesis of GBM cells spreading along different spatial trajectories. The dataset includes gene expression data from 41 patients taken from the following anatomic regions (with well-characterized histo-pathologic features) present in GBM: cellular tumor (CT), leading edge (LE), infiltrating tumor (IT), pseudopalisading region around necrosis (CTpan), and microvascular proliferation (CTmvp) (see Fig. 1a–b, Supplementary Data 1). All patients received standard therapy: resection, radiation, and chemotherapy. In agreement with the analysis of Puchalski et al.[10], clustering of the samples based on their gene expression profiles reveals location-specific differences (see Supplementary Fig. 1).

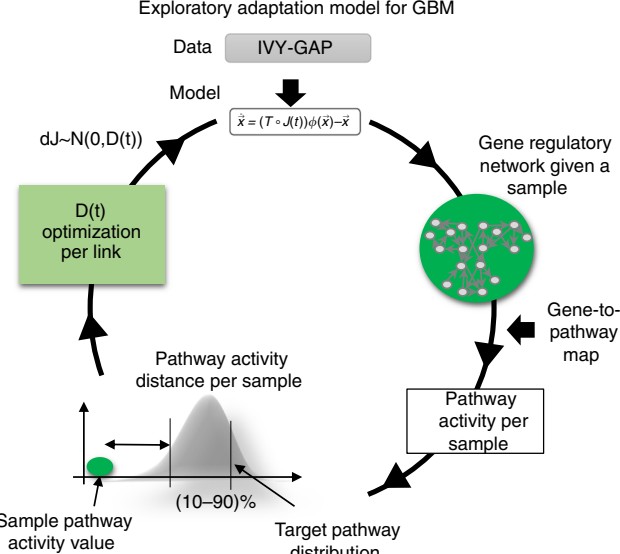

Exploratory adaptation model for GBM

**Fig. 2 Flow chart of single-sample dynamics model.** At every time step $t$, the model performs the following steps: (1) the vector of gene expression ($x(t)$) is updated based using the gene regulatory network, with edges weighted by correlations between the expression of the nodes genes; (2) the phenotype ($y(t)$) is calculated as the pathway activity vector based on the gene expressions $x(t)$; (3) the minimum distance between the current sample's phenotype and the 10–90th percentile interval of target's phenotype distribution is estimated for each pathway; (4) If the distance is 0, that is, the sample's phenotype is within the target's phenotype distribution, the exploratory capacity, $D(t)$, decreases relative to its previous time value. This process is repeated until the phenotype converges to its target, or the simulation is stopped at $t = 1000$.

**GBM spatial phenotypic trajectories.** We constructed a set of possible spatial trajectories using the shortest phenotype distance. We define a phenotype at the functional activity level. We define a sample's phenotype to be the vector of its pathway activities; we use KEGG ontology[15] for pathway definitions ($n = 182$) and compute the pathway activities based on the expression of the focused gene list (Fig. 1c). We define a location's phenotype to be the vector of pathway activity distributions of all the samples from that location (Fig. 1d–g).

Comparison of gene-expression based clustering to pathway-activity based clustering (see Supplementary Fig. 1) reveals that location-specific differences are robustly captured at the pathway-activity level: distinct patterns of activity can be observed for each location. These differences are, however, more muted than at the gene-level, reflecting that pathway activities are less sensitive to perturbation of individual genes and the redundancy and compensatory capabilities of cells. Distinct patterns of variation in location phenotypes are also apparent: CT and IT show higher variance in the activity distributions of a number of pathways, and similar patterns of variations compared to the other locations (see Supplementary Fig. 1).

To quantify phenotypic changes due to adaptation we defined two measures of phenotype distance: Pathway Distribution Distance (PDD) (defined as the vector of distances between individual pathway distributions and measured using Euclidean distance), and a coordinated Global Distribution Distance (GDD) (defined as the summation of PDD between a pair of phenotypes) (Fig. 1d–g, Supplementary Data 1). We assessed the statistical significance of GDD using permutation tests (see methods, Supplementary Data 1). We also estimated the significance of pathway activity variations between two locations using unpaired, two-sided $t$-test (BH-adjusted $p$-values) and found good correlation with the PDD values (Pearson's coefficient 0.5–0.7, see Supplementary Data 1 and Supplementary Fig. 1).

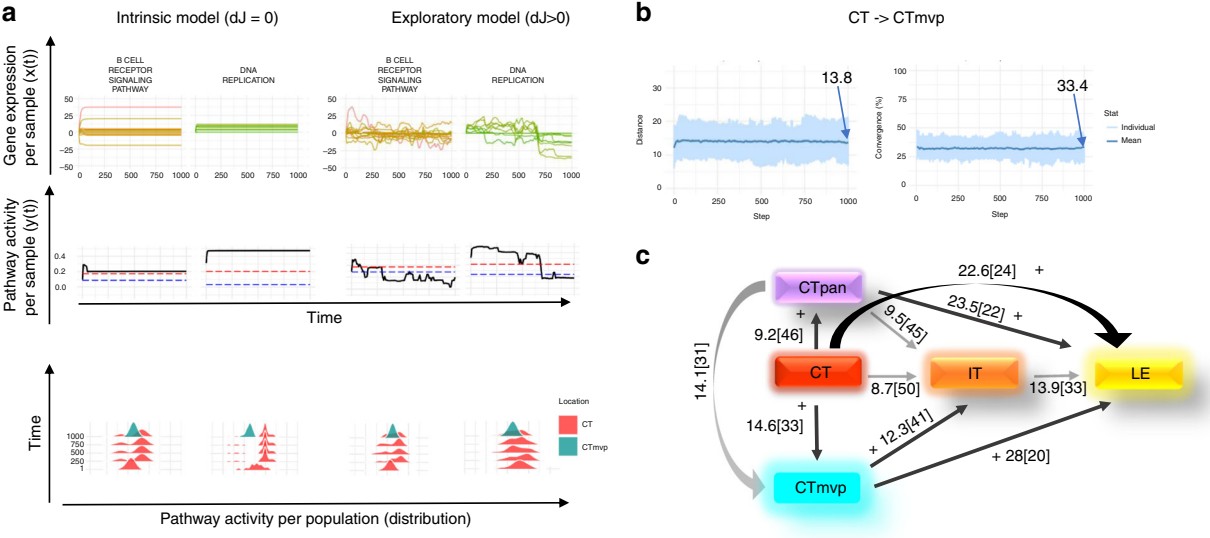

**Fig. 3 Cellular response to known and new environmental stimuli. a** An example depicting the phenotype transition CT→CTmvp: first, for a given sample changes for two example pathways (DNA replication and B cell receptor signaling pathways) and their corresponding genes were plotted over time. The lower panel show the activity distributions of the two example pathways obtained from the set of samples from each location. Simulation showed that the exploratory behaviors of both pathways differ from the intrinsic behaviors, therefore indicating that new adaptive pathway activities are due to the exploratory ability. **b** Following the calculations in sub-plot A, global distance per sample, together with the number of converged pathways were calculated over time. Light blue color represents individual samples' temporal phenotype distances, and the dark blue line represents the mean value of that global distance. Initial $D$ ($t = 0$) was set to 0.1, and the distance ($t = 1000$) reached 13.83, while number of converged pathways was reached to 33.4% on that time step. **c** Simulation results of the updated location-based phenotypes were re-calculated to include exploratory capacity ($dJ > 0$). Black color with a plus sign denote weights that are different from those of the intrinsic behaviors ($dJ = 0$) (intrinsic behavior is marked with gray lines).

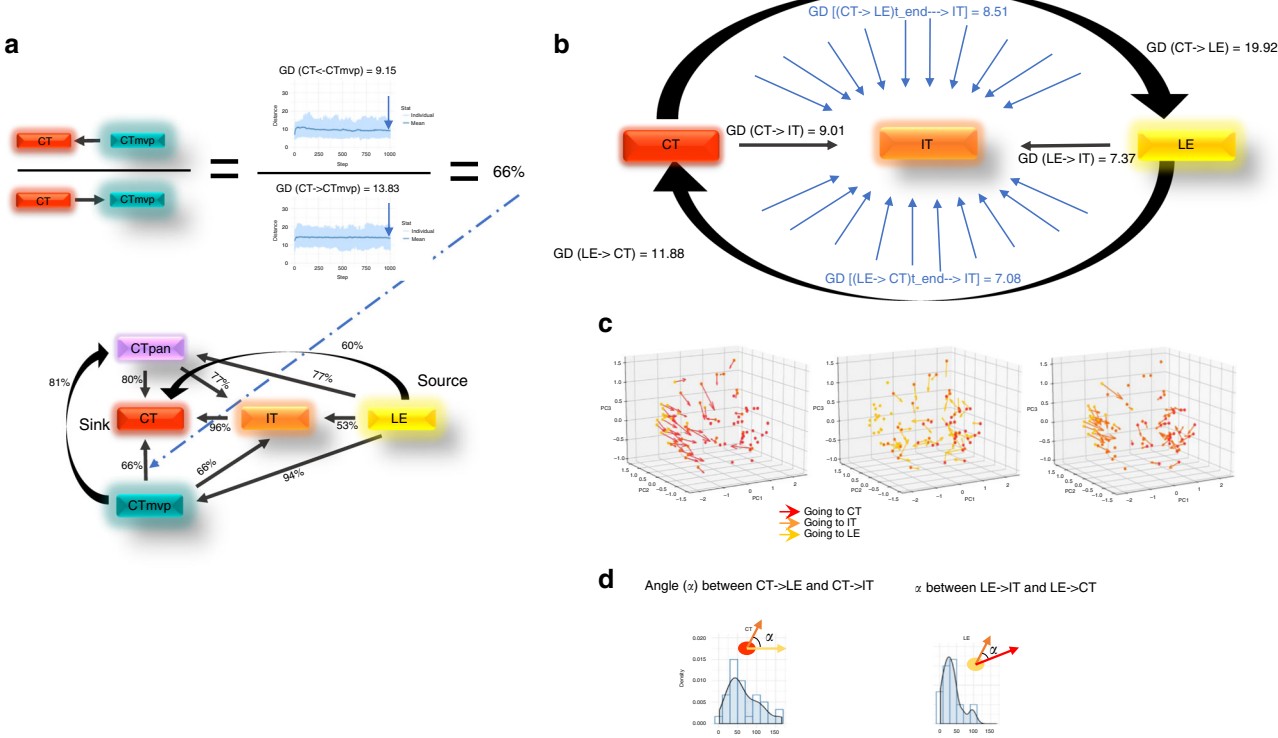

**Fig. 4 Properties of the simulated phenotype dynamics. a** The reverse process of transitioning between phenotypes was simulated for each edge of the location-based phenotype network from Fig. 3 (as seen by the reversal of the directions of the edges of the network in Fig. 4 compared to those of the network in Fig. 3). The ratio of global distance differences, and the ratio of number of converged pathways in square brackets, between the two directions are shown as weights. **b–d** The importance of spatially intermediate phenotype states. Focusing on the trajectory of CT→IT→LE, and the reverse trajectory, we examined three characteristics of the resulting intermediate phenotypes and the reversal process: **b** the ratios of the global distances between the two paths show that cells reverting their trajectory need smaller changes in phenotype to get back from LE to CT, compared to reaching LE from CT using the original path: they are closer to the target phenotype when reverting (11.88 distance) compared to the original end point (19.92); **c** vector fields of the simulated phenotype dynamics were plotted using PCA; the arrow origins are the starting phenotypes ($t = 0$) and the end points are the simulated phenotypes ($t = 1000$). The color of the dots corresponds to the starting phenotype location, whereas the colors of the arrows depict the target phenotype; for example, red arrows that start from a yellow dot are simulation results of a case where the initial condition is LE, and the target phenotype is CT; D) the angles between the original paths of CT→ LE and CT→IT, and angles between the reversal paths LE→CT and LE→IT are narrow, which indicates a constrained sphenotype exploration space.

The GDD, PDD, and differential pathway activity reveal a spatial pattern with CT in its center, and LE as the farthest phenotype. CT's heterogeneous pathway activities partially overlap with those of IT, CTpan, and CTmvp. The spatial organization is also apparent in a Between Group Analysis (BGA) plot (Fig. 1d–g). The shortest phenotype distance (using GDD, PDD, and number of differentially active pathways) exposes three well-defined spatial trajectories, two of which relate to oxygen supply (Fig. 1h–i): (1) CT→ CTpan (moving toward hypoxic regions), (2) CT→ CTmvp (moving toward oxygen-rich regions), and (3) CT→ IT→ LE (spreading throughout the brain). The pathway activity BGA plot displays overlap between locations, suggesting a continuous transition along the spatial trajectories; the overlap is less visible in the gene-level BGA (see Supplementary Fig. 1). Lastly, we observed that the span of trajectory of CT→ IT→ LE is the largest (in both values) compared to the oxygen related trajectories (CT→ CTpan and CT→ CTmvp). Moreover, locations CT, IT, and LE are closer to CTpan than to CTmvp.

The static snapshot of the system shows that smaller phenotypic changes are required for CT cells to adapt to the conditions in the locations of CTpan (hypoxic regions) or IT (infiltrating) whereas adapting to CTmvp area (oxygen-rich regions) requires larger changes. CT cells that have adapted to an IT area require smaller additional changes to further adapt to the

LE area. Yet, such additional changes create a larger step between CT and LE; similarly, for CTpan and LE or CTmvp and LE.

**Exploratory adaptation model for GBM.** We next studied the dynamical process of adaptation along the identified spatial trajectories by building on the theoretical model proposed by Brenner et al.[4]. The theory of exploratory adaption stipulates that adaptation is achieved through small random changes in the gene regulatory network that eventually give rise to a range of new steady phenotype states. We investigated how these concepts apply to GBM.

Microenvironment stimuli are well characterized to be non-uniformly distributed and gradient-based[16,17]. In a 3D tumor mass, a gradient of oxygen or hypoxia is established by the combined effects of cellular metabolism and oxygen diffusion; this gradient-based nature of the microenvironment stimuli, is in fact, a major barrier to recapitulating the in vivo conditions in in vitro models[18–20]. The Blood Brain Barrier (BBB)/Blood Tumor Barrier (BTB) introduce further osmotic gradients. Therefore, uniform response may not be a biologically-relevant assumption for GBM. Moreover, different stressors (nutrient depletion/hypoxia/osmotic pressure) may perturb different mechanisms and pathways, making it impossible to fully capture phenotypic changes with a single global compression value. We therefore coupled global measurements with more detailed

pathway activity level changes. We incorporated our definition of pathway-driven phenotype and introduced a notion of steady state convergence–a sustained phenotypic state that the cells may reach through exploration (Fig. 2).

We developed a biased exploratory adaptation approach that models how reaching phenotypic states that are compatible with the novel environment may reduce the biological pressure to continue exploration. We model a sample's exploratory process as being driven by stochastic changes in the regulatory network with a scale parameter ($D \sim N(0, D\_\text{max})$); the magnitude of the change may decrease over time ($D\_\text{max} \geq D(t)$) as a function of the pathway vector distance and associated convergence criteria. At each time step, an exploring sample's phenotype is computed as the vector of its pathway activities; a pathway whose activity is in the 10–90% interval of the target pathway distribution is considered to have converged. The magnitude of the stochastic perturbation to the regulatory edges targeting genes belonging only to converged pathways is from then on decreased (which is achieved by reducing $D(t)$). See methods for details.

**Established responses to familiar environmental stimuli.** Before considering adaptation to novel challenges we asked how cells behave in a familiar local environment. Establishing this baseline behavior enables estimating how responses to new stimuli differ from this pre-evolved, intrinsic behavior, which we consider the control behavior. Starting from the static location-based phenotypes and our GBM model, we estimated the expected cellular phenotype over time in a fixed homogeneous environment, in which the strength of regulatory edges ($J$) is constant ($dj = 0$). In this setting, the cell acts based on a pre-evolved molecular system with conserved feedback loops to execute a set of behaviors under no external stress conditions, which is modeled as lack of stochastic perturbations to the regulatory network. The simulation predicts that cells in all locations can intrinsically evolve their phenotypes over time to phenotypes with similar expected global distance from their initial phenotypes. The location phenotypes can be ordered from the most intrinsically rigid to the most intrinsically-plastic as follows: CTpan, IT, CT, LE, and CTmvp (we represent the degree of plasticity as the width of the aura in Supplementary Data 2, Supplementary Fig. 2, and illustration in Fig. 3c). Thus, CTpan displays the smallest phenotypic changes from its initial phenotype until reaching a stable phenotype, whereas CTmvp displays the largest intrinsic changes. All differences between initial and final time point intrinsic phenotypes are statistically significant (*p*-values were estimated using permutation tests of GDD; see methods and Supplementary Data 2). By embedding and visualizing these simulated data over time using Principal Component Analysis (PCA) and BGA we observed that the three main trajectories are preserved under familiar conditions (Supplementary Fig. 1). We also analyzed the differential activation of the pathways due to the intrinsic behaviors (by calculating PDD and unpaired, two-sided *t*-test, with strong agreement between the two Pearson's coefficients ~0.7) (see Supplementary Data 2, Supplementary Fig. 2). For example, pathways that undergo changes in activity due to intrinsic network dynamics are: TGF-beta signaling for all locations except LE, JAK-STAT signaling for LE and CTmvp, T cell receptor signaling for all locations except CT, and WNT signaling for all locations.

Additionally, by simulating the intrinsic phenotype at convergence (with the phenotype remaining sData after 1000 time points), we estimated the phenotype distance from the simulated phenotype to other location phenotypes as targets. Our results (see weighted network in Supplementary Data 2) show that several directions are intrinsically closer in phenotype compared

to others (from closest to farthest): CT→ IT, CT→ CTpan, and CTpan→ IT, whereas the largest intrinsic distance is that between CTmvp→ LE.

**Novel responses to new environmental stimuli.** The physical distance and the differences in environmental stress between locations subject cells to both known and unforeseen stimuli. We investigated whether our model could explain the dynamical process of transitioning between two phenotypes along the spatial trajectories. For a given location pair, for example CT→CTmvp, we simulated for each sample *s* from CT, the phenotypes that can be obtained through exploratory adaption over a given time interval [initial, final]. We computed the distance of *s*'s phenotype at time point final from CTmvp (as the sum of the distances of the pathway activities of *s* from the 10–90% percentile of the target CTmpv pathway distributions; see methods). The degree to which the samples of a location reduced their distance to their target phenotype measures the degree to which exploratory adaption is responsible for the differences in phenotypes. We compared these distances with those obtained from simulation of intrinsic behavior (dj = 0), and when these distances were not smaller than those obtained in the intrinsic case, we concluded that exploratory adaption is not necessary to explain those phenotype differences.

These results are depicted in Fig. 3c. Transitions that require some degree of exploratory adaption include: CT→LE, CT→CTpan, CTpan→LE, CT→CTmvp, CTmvp→LE, CTmvp→IT. Focusing on the CT→CTmvp case, we examined in detail two GBM relevant pathways, DNA replication and B cell receptor signaling; these are illustrated in Fig. 3a–b and in animations in Supplementary Movie 1 (intrinsic animation) and Supplementary Movie 2 (exploratory adaptation animation). Simulation results showed that both pathways behave differently due to exploratory adaption compared to the intrinsic, control case. For example, DNA replication activity transitions from a multimodal distribution to a unimodal distribution, which converges to the target pathway distribution. There is no difference in the distances of CTpan→IT between the exploratory and intrinsic behaviors. Interestingly, the phenotypic differences between the simulated CT→IT and IT→LE are close enough; in fact, added exploratory abilities increase the final distances compared to the system with no noise. In all these cases, a cell does not need the exploratory part in the adaptation process to explain its phenotypic differences but rather uses its intrinsic system. In summary, microenvironmental pressures primarily destabilize the source node CT and drive exploratory phenotypic changes to the more stable or sink node LE. We performed permutation tests to estimate the significance of the GDD between the intrinsic simulated behavior of the starting location at *t* = 1000 compared with the EA simulated behavior at *t* = 1000. All *p*-values were significant (see methods and Supplementary Data 2).

Recognizing that distinct stimuli characterize the environment of distinct locations we also explored whether location-specific levels of stochasticity may be affecting the cell's decision-making and impacting its regulatory network. We tested a range of such exploratory variabilities ($D$= 0.01, 0.1, 0.5) and found that the resulting phenotypes, while slightly noisier for larger values of $D$, are on average similarly affected (see Supplementary Data 2).

**Reverse adaption.** We next investigated the ability of cells that have succeeded to adapt to novel environmental challenges to revert their phenotypes: could LE cells, for example, revert their phenotypes to the original CT phenotype? Would exploratory adaption be necessary to achieve these phenotypic changes? To

address this question, for each original exploration (for example, starting from CT with target CTmvp, with resulting GD (CTth rvp) = 13.83) we simulated the reverse exploration (starting from CTmvp with target CT, with resulting GD (CT ← CTmvp) = 9.15) (see Fig. 4a, Supplementary Data 2). We found that the reverse distances of all locations back to CT were smaller than the distances of phenotypes leaving CT toward other locations. This suggests that the initial adaption of a cell leaving the CT phenotypic state requires more changes than relapsing back. LE is the phenotype that is most capable of reverting to any of the other phenotypes. Altogether, in this mirror directed adaptation network, CT emerges as a sink (having global minimal distance) more readily reachable by all the other phenotypes; IT and CTpan act as secondary sinks, and LE and CTmvp as sources. We also assessed whether exploratory adaption is necessary for this reversal by simulating process reversal without stochastic perturbations of the regulatory network. We found that this process is more efficient, leading to states that are closer to the target phenotypes regardless of the start location. We concluded that reverting cells utilize the plasticity that they have internalized, rather than exploratory adaption. Permutation tests were performed to estimate the significance of the GDD between the intrinsic simulated behavior of the starting location at $t = 1000$ compared with the reversal EA simulated behavior at $t = 1000$. All p-values except for LE→CT were significant (see methods section, Supplementary Data 2).

**The importance of spatially intermediate states**. We showed that some degree of exploratory adaptation could explain the changes in several location-based phenotypes. When the gene-regulatory network is affected by stochastic perturbations we anticipate that multiple possible evolutionary paths could lead to new adapted states. We next asked whether the process of exploration is constrained by the environment. For example, are cells starting from CT and moving toward the targeted distant location of LE constrained to phenotypes similar to the intermediate IT? To address this question, we estimated the distance of phenotypes reachable by CT cells transitioning toward an LE phenotype. At each time point, we also calculated the distance to the intermediate phenotype IT (Fig. 4b, blue arrows). We compared the final distance resulting from moving from CT with target IT (that is, GD (CT isr = 9.01) at time point $t = 1000$) with the final distance of the phenotype obtained from moving from CT toward LE with respect to IT (GD (CT → LE) → CT → = 8.51 at time point $t = 1000$).

We depicted the resulting simulations in a 3D PCA plot (Fig. 4c) where red, orange, and yellow dots represent phenotypes of samples from CT, IT, and LE, respectively. Arrows represent the trajectories of individual samples and connect the starting phenotype to the simulated phenotypes. Both the population of arrows CT→IT and IT→LE, as well as the population of arrows LE→IT and IT→CT reveal alignments that suggest that transition along CT↔LE is constrained to phenotypic ranges approaching the intermediate phenotype IT. To quantify the degree of such alignment we computed the angles between the arrows from the same origin for all arrows starting from CT: CT→LE and CT→IT, and the angles of the arrow pairs LE: LE→IT and LE→CT. First, we found that the mean angles are narrow, confirming the alignment observed from the PCA. Second, we found that the distribution of LE→IT and LE→CT pair angles (Fig. 4d) has a smaller mean and range than that of the CT→LE and CT→IT distribution. Therefore, process reversal along LE→CT is more constrained, and less exploratory than CT→LE; this is in line with our previous finding that intrinsic plasticity suffices for phenotype reversal.

Interestingly, while at the population level the alignment of the arrows is along the main CT↔IT trajectory, at the individual arrow level (which represent the individual sample behavior) we observe orientation toward the nearest samples of the target distribution. That is, rather than being driven to resemble the mean of the target population, the individual samples seek to resemble the nearest samples from the target distribution.

**A therapeutic roadmap**. The intermediate phenotype IT, better characterized as a continuum of phenotypes, has emerged as a critical and necessary step while transitioning along the CT↔LE axis. We next asked whether discretizing the continuous phenotype could serve as a therapeutic roadmap. We asked whether isolating and decreasing the variability of this phenotype (through removal of some IT samples from that population) would slow the phenotypic flow along the trajectories of CT↔LE, and thus impact the ability of CT to transition to LE and back. We focused on the IT↔LE portion of the trajectory, and compared the transition capabilities between IT and LE by performing simulations on sub-populations of IT and LE constructed as follows: (1) removed 3 (from IT) + 2 (from LE) closest samples to the other phenotype, (2) removed 4 (from IT) + 3 (from LE) closest samples to the other phenotype, and (3) removed 4 (from IT) + 3 (from LE) randomly selected samples. From our analysis (see Supplementary Data 2) we observed that the more homogenous and distant the subpopulations are (the sub-populations with 3+4 closest samples taken out being the extreme) the largest the final (post-adaptation) phenotypic distance remains. By comparison, randomly removing the same number of samples from the sub-populations, or removing fewer of the closest samples, results in smaller final differences.

**Immune signatures of the phenotypes**. We considered the question of how location-specific immune signatures, as opposed to functional pathway activities, would respond to micro-environmental challenges, and whether intrinsic and exploratory adaptation could bridge any differences in such signatures. We used xCell[21] to deconvolute the (whole genome) bulk expression profiles of the different Ivy-GAP locations, and obtained the immune signatures plotted in Fig. 5. The most enriched micro-environment appears to be around microvascular proliferation, which suggests that the BBB/BTB is leaky: we see endothelial signatures, but also fibroblasts, pericytes, and some enrichment of immune signatures. This suggests that there is trafficking of immune cells in the microenvironment, but the paucity of these cells in other regions of the tumor (as seen by the weak signal from other locations) suggests that few resident or trafficked immune cells are present in the other regions. We, therefore, did not pursue the question of adaptation of immune signatures further. However, the results are in agreement with GBMs' being mostly immunologically cold tumors: they arise in the immunologically privileged BBB-protected environment of the brain, and secrete factors that inhibit both the innate and adaptive immune systems. Nevertheless, due to the poorly-formed neo-vasculature of GBMs, the BBB and BTB can often be leaky, which leads to the level of immune cell infiltration reflected in Fig. 5.

## Discussion

Obtaining large scale time series molecular profiles of patient tumors is next to impossible in the GBM context as this would require repeated invasive procedures within a typically short period of time; on the other hand, animal models do not recapitulate the patient profiles due to significant differences in space scale. Here, we offer a unique modeling approach that enables inferring temporal aspects of the processes from sparse but

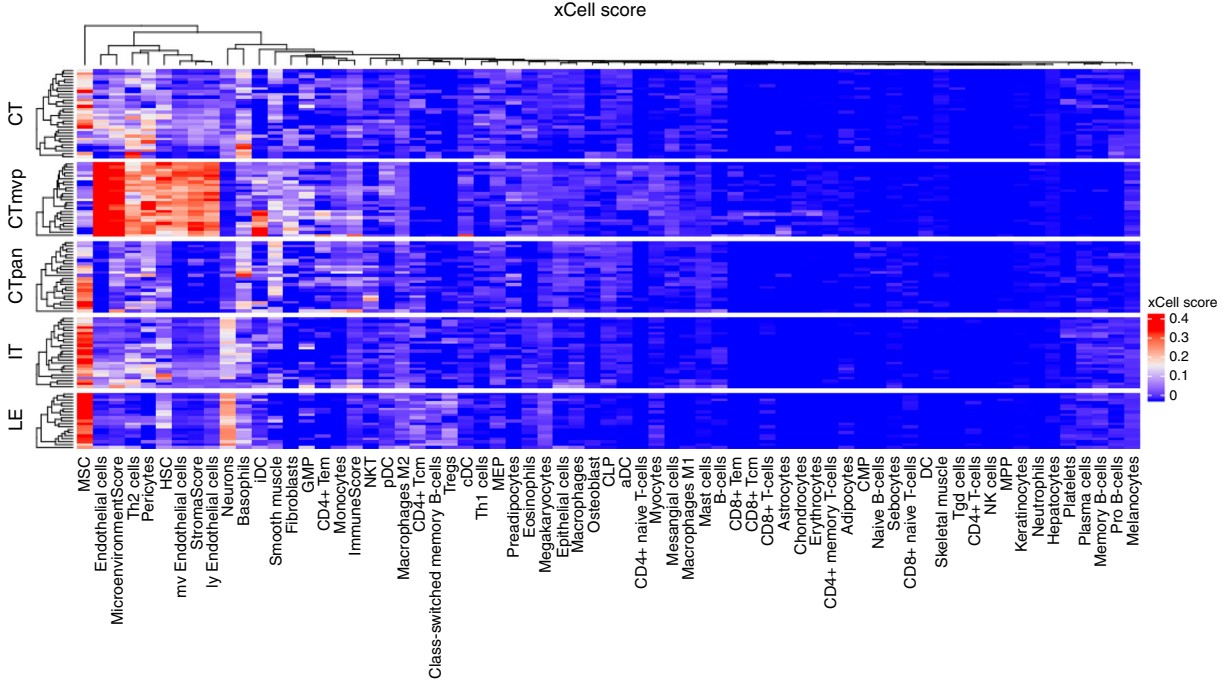

**Fig. 5 Immune signatures of the phenotypes.** xCell[13] was used to deconvolute the (whole genome) bulk expression profiles of the different Ivy-GAP locations, and obtain the plotted signatures.

obtainable patient spatial information. This work has clear implications for understanding the GBM recurrence dynamics, and broader translational impact for heterogenous solid tumors. The question of how to effectively target heterogeneous tumors is crucial for most cancer therapies: adequately sampling, modeling, and predicting response to therapy are especially challenging for such cancers. These challenges are multiplied for cancers capable of invading and adapting to new environmental stressors, as these processes include not just complex dynamic systems, but also incorporate stochasticity and uncertainty. In this study, we addressed the question of predicting varying tumor heterogeneity over spatial trajectories by translating each location's transcriptomic profile to a phenotype, and examined the transformation between the location-based phenotypes. We proposed several extensions to the exploratory adaptation model that enabled us to infer temporal behavior starting from the Ivy-GAP spatial information. We introduced new notions of phenotype, and phenotype trajectory, and proposed a model of gradual adaptation. We demonstrated that exploratory adaptation can explain some extent of the transformation of several locations' phenotypes to others: CT→CTpan, CTpan→LE, CT→CTmvp, CTmvp→LE, CTmvp→IT, CT→LE. Cells that gain sufficient plasticity no longer need such exploration to transform their phenotypes, or revert them to the original ones.

Another insight obtained from our study is that exploratory cells may search and settle for the nearest sufficiently adapted phenotypes (reaching local optima of semi-stable phenotypes) rather than the best fit for that location (the mean phenotype distribution of that location). This results in intermediate phenotypic continua that are readily capable of surviving new challenges. Blocking formation of intermediate semi-stable phenotypes and in effect isolating each location-driven phenotype may disrupt the cells' ability to survive through exploratory adaptation. Gradual adaption as modeled by our phenotype search and convergence procedure has also been observed experimentally: Carmona-Fontaine et al.[22] showed that alterations in metabolism of cancer cells create predictable gradients of

extracellular metabolites in response to gradients of hypoxia in the microenvironment, which, in turn, further orchestrate the phenotypic diversity of tumor cells. In addition, the analysis of reverse trajectories could imply that while learning a novel phenotype, the previous phenotype remains an attractor. Thus, adaptation could occur along focused functional pathways, necessary for coping with the stimuli, rather than globally.

How could our findings be used to hinder the ability of LE tumor cells to reform a new CT? Reducing the likelihood of relapse necessitates better understanding and disabling of the steps involved in recurrence: how isolated tumor cells survive and adapt to new microenvironments, as well as how they reacquire phenotypic features that enable formation and growth of new tumors (for example, as in LE→CT). We addressed these fundamental questions and showed that GBM cells can change phenotypes under environment pressures using intrinsic or exploratory adaption depending on their initial molecular state, and physical location. We demonstrated that intrinsic adaptation is sufficient to bridge short phenotypic distances as in the case of CT→IT, IT→LE; by contrast, bridging larger phenotypic distances such as CT→LE necessitates exploratory adaption. We also showed that exploratory adaption imbues phenotypes with sufficient plasticity to revert to the original phenotypes. However, when transitioning along the CT↔LE trajectory, the intermediate phenotypes are constrained to resemble the IT phenotype. Therefore, we propose to target new tumor formation (after GBM resection and standard therapy) by targeting the IT-phenotype distribution; we hypothesize that this approach of targeting critical points in phenotype trajectories would reduce the likelihood of LE tumor cells to reform new CT tumors.

## Methods

**Reproducibility**. Throughout the study we attempt to use standard, well-established bioinformatics approaches in an effort to increase the reproducibility of the results and the generality of our framework.

**Preprocessing and differential gene expression analysis**. We selected genes relevant to gliomas by analyzing expression of low-grade gliomas (LGG) and GBMs

from TCGA patient cohorts. RNA-Seq mRNA expression data were downloaded in 08/03/2018 using R's[23], TCGAbiolinks library[24]. In all, 672 primary glioma samples (516 LGGs, 156 GBMs) were preprocessed, normalized, and filtered as recommended in R's TCGAWorkflow[25] and submitted to differential gene expression analysis. In all, 2975 genes passed the filtering criteria of Fold Change ≥2 and −log10 False Discovery Rate ≥5. 1441 genes from the upper 10% quartile of variation in the combined glioma cohort were also added to the selection.

**Regulatory network.** Human Transcription Factor (TF) -Gene regulatory network data were downloaded from RegNetwork[14] (http://regnetworkweb.org/) on 06/2017. RegNetwork integrates the curated regulations in various databases and the potential regulations inferred based on the transcription factor binding sites. We augmented our glioma focused gene list with any TFs that target them, resulting in a list of 4121 genes. The resulting GBM TF-Gene regulatory network consists of 4121 nodes and 78,686 edges, and average degree 19.09.

**Ivy-GAP data.** RNA-Seq profiles for 270 laser-microdissected samples from 41 GBM tumors and sampled from different anatomical regions of the tumor were downloaded from the Allen Institute's Ivy Glioblastoma Atlas Project (Ivy-GAP) (http://glioblastoma.alleninstitute.org/, GSE107559)[10]. These are given as fragments per kilobase per million (FPKM), and further adjusted with TbT normalization (by scaling each sample based on the summed expression of all genes that are not differentially expressed). In all, 122 samples from the Anatomic Structures (AS) study, obtained from 10 tumors were further selected for analysis. The locations (L) and number of samples are as follow: cellular tumor (CT, 30 samples), leading edge (LE, 19), infiltrating tumor (IT, 24), microvascular proliferation (CTmvp, 25), pseudopalisading cells around necrosis (CTpan, 24). The samples per tumor and their locations are summarized in the Supplementary Data 1.

**Gene-to-pathway map.** Annotations for (186) KEGG[15] pathways were downloaded from Molecular Signature Database 7 V6.1 (collection c2.cp.kegg.v6.1. symbols)[13].

**Pathway activity per sample.** Gene set enrichment (GSE) analysis enables condensing information from gene expression level to functional pathways or signatures summary, thereby reducing the noise and dimensions of the data. Such functional pathway summary, also known as pathway activity or pathway enrichment score, captures the degree to which the genes involved in a pathway are coordinately up or down regulated. Many GSE methods are supervised and population based, in that they compute enrichment scores as comparisons between two groups of the population (for example, case/control). However, our simulation approach, tracking changes at a single-sample level, necessitates the use of an enrichment approach that can be applied at the single-sample level, and without regard to the class labeling. ssGSEA is a non-parametric, unsupervised method, which starts by evaluating the rank of each gene per sample; for each pathway/gene set a Kolmogorov-Smirnov-like rank statistic is calculated, giving an estimate of the overall rank of the genes belonging to that pathway compared to the rank of the rest of the genes[26]. Hänzelmann et al.[27] compare several competing GSE methods, including ssGSEA and GSVA, and find that these two outperform the others both with respect to sensitivity and accuracy of the methods to identify differential pathway activity. R's GSVA[27] library was used to compute the pathway activity for each sample using method ssGSEA, from Ivy-GAP data or simulated data. Activities for 182 of the pathways could be computed, based on the expression of the GBM focused genes.

**Location-driven phenotype distributions.** We assume that there exist phenotype distributions at the location level across patients for the following reasons. First, the Ivy-GAP dataset contains multiple samples from the same tumor for some of the regions, while no samples for others; so multiple possibilities and sometimes none exist for mapping these samples into initial-final pairs per patient. Second, GBMs have been shown to exhibit intra-tumoral heterogeneity at the single cell resolution; Patel et al.[28], for example, showed through single cell transcriptomic profiling that primary GBMs display a mixture of the four established GBM transcriptomic subtypes, with individual cells from the same tumor displaying one of the four subtypes[29]. Indeed, as seen from the molecular characteristics of the tumors presented in the summary table above, some of the tumors display a mixture of transcriptomic subtypes (Classical + Mesenchymal, or Classical + Neural, for example), confounding the question of which of the samples moved to a different location. Last, the location-centric distributions capture the location-specific phenotypes, which as shown both in the Ivy-GAP landmark paper as well as our Supplementary Figures (pathway heatmaps) are quite distinct.

**Pathway Distribution Distance.** For each location, and each pathway, a pathway activity distribution can be computed from the pathway activities of the individual samples. For a pathway ($j \in 1..m$) and two locations $i, k \in \text{Location}(L)$, the Pathway Distribution Distance (PDD) $\left( \text{PDD}_{j, L(i) \to L(k)} \right)$ estimates the distance between $j$'s activity distributions for the locations i and k; the distances were

computed as the Minkowski distance with Euclidean option (see Supplementary Data 1) using R's HistogramTools[30] library.

**Global Distribution Distance.** The phenotypic differences between any two locations i, and k are captured using the global distance ($\text{GDD}_{L(i) \to L(k)}$) between the two locations computed as the sum of the PDDs over all pathways: $\text{GDD}_{L(i) \to L(k)} = \sum_{j=1}^{m} PDD_{j, L(i) \to L(k)}$, $m = 182$ (see Supplementary Data 1). Permutation tests were used to estimate statistically significances between: (1) initial GDD from data, (2) simulated results of intrinsic initial versu final time points for all locations, and (3) between simulated results of intrinsic versus EA in final time points for all locations. The normalization step was achieved by mean subtraction of each initial distribution, and the randomization part of the permutation test required to overlap samples with 0.1–0.9 of their original distributions. All p-value results are summarized in Supplementary Data 1–2.

**Differential activity (DA) analysis.** The most significant pathway activity differences between any two locations, were estimated using unpaired, two-sided t-tests; a BH-adjusted p-value cutoff of 1e-06 was used as threshold of significance (see Supplementary Data 1 and 2).

**Model.** We introduce our expanded model of exploratory adaptation using patient-derived GBM data from spatially anatomical structures. In addition to the details of the original Brenner's model[4], we describe here our expanded model. The intrinsic model describes the time evolution gene expression based on their gene regulatory network topology and their interactions strength. The model consist of initial focused GBM network, $X = (x_1, x_2, \text{two} x_n)$, $n = 4,121$, governed by the following nonlinear equation: $\dot{X} = (T * J)\varphi(X) - \beta X$ ; where the adjacency matrix $T$ is the corresponding human TF-Gene network with 78,686 edges in binary values (0/1); $J$ is the Pearson's correlation matrix specifying the actual interaction strengths based on the Ivy-GAP samples data per spatial location; * is an element-wise (Hadamrd) product; $\varphi(X) = \tanh(X)$ is an element-wise saturating function; time scale was performed with $\tau = 4t$; and $\beta = 0.5$ the constant vector of relaxation rates.

Our definition of phenotype, $Y$, is a vector of cell's functions (KEGG pathways), $Y = f(X)$, $Y = (Y_1, Y_2, \text{hw} Y_m)$, $m = 182$. The phenotype value ($f(X)$) depends on gene-to-pathway map and pathway activities based on gene expressions (from simulations or data) per sample. Each pathway distance (PD) per sample, is calculated based on the distance between the actual single-sample pathway activity compared with the target activity 10–90% distribution percentile. To estimate the distance in phenotype between a sample and a target phenotype, we define a phenotype distance (i.e., global distance (GD)), which summarizes the pathway activity distance (PD) across all pathways. Convergence to specific phenotype was reached if all simulated pathway activities were within the range of 10–90 percentiles of the target phenotype destination. Note, in simulations, the phenotype, pathway distance (PD), and phenotype distance (GD) are calculated per sample, and not between distributions (as calculated before in the data-driven step: PDD and GDD).

To model the exploration dynamics in phenotype due to new environment stimuli, the intrinsic model incorporates additional stochastic process. As in Brenner's model, we also allowed small random changes in the interaction strengths of the regulatory network, forming a random walk in the elements of the correlation matrix $J$: $dJ_t = \sqrt{D \cdot M(y_t, y^{\text{Target}})} \cdot dw_t$, where the $w_t$ follows the Wiener process: $w_t \sim N(0, 1)$, $M$ is the phenotype convergence function:

$M(y_t, y^{\text{Target}}) = \begin{cases} 1, & y_t \in y^{\text{Target}} \\ 0, & \text{otherwise} \end{cases}$ , $y_t$ is the temporal pathway activity vector:

$y_t = (y_{1t}, y_{2t}, \ldots y_{182t})$, and the target distribution vector $y^{\text{Target}}$ is the interval [10–90th] percentiles of the target phenotype. However, this process has no longer uniform scale parameter, $D$, but may change per interaction strength. As a result, $D$ and $dJ$ have now the same dimension as the matrix $J$, that is $4121 \times 4121$. We introduce a biased exploratory adaptation model to explore high dimensional search space with biological and functional relevance. Our approach includes an exploratory process that starts with an initial stochastic noise scaled with parameter $D_{\max}$, $D(t = 0) \sim N(0, D_{\max})$, which changes over time ($D_{\max} \geq D(t)$) as a function of the pathway vector distance. For every given time step, a pathway distance vector is calculated between the current temporal pathway activity vector to the 10–90th percentile of the target phenotype distribution. This process is independently executed for every pathway. In a case where a pathway converges to its destination (that is, its activity is within the 10–90th percentile of the target distribution), all member genes of that pathway that do not belong to other pathways that have not converged yet, will decrease their random exploration from the next time point on. So if a gene $k$ belongs only to pathways that have converged, then $D(k, :)_{t = i+1} = 0.95 \cdot D(k, :)_{t = i}$, and the same factorization on the corresponding columns. In this way, a cell could first change the pathways most relevant to the present external stimuli, and subsequently gradually stabilize the remaining pathways to a phenotype fit for that spatial location.

In all simulations, unless otherwise specified, the last time point was set to $t = 1000$. For simulations that include exploratory capacity, initial $D$ ($t = 0$) was set to 0.1. Results of GD values for simulations using $D = 0.01$, and $D = 0.5$ are shown in Supplementary Data 2.

**Notes about figures**. Plots were produced in R using ggplot2, and Python using matplotlib. Between group analysis (BAG) with Correspondence Analysis option was performed using R's made4 library[31]. Top 100 genes on the first three principal components were used to create heatmaps of (zscored) expression. Correlation between PDD (or PD) and DA were calculated using R statistics library, with method Pearson. Python's scikit was used to obtain the PCA coordinates, and matplotlib for visualizing the vector fields.

**Reporting summary**. Further information on research design is available in the Nature Research Reporting Summary linked to this article.

## Data availability

All datasets that were used and support the findings of this study are available online. The data (processed gene expression, phenotype labels, gene lists, pathway ontology, computed pathway activity, and TF-gene regulatory networks), and our code can be downloaded from our GitHub repository (https://github.com/oricel/PhenoExploreR, or https://github.com/OritLavi/PhenoExploreR).

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

## Acknowledgements

O.L. would like to thank Drs. Michael M. Gottesman and Tom Misteli (NCI, NIH) for their comments and support. We also thank George Leiman (NCI, NIH) for editorial assistance, and Erina He (NIH Medical Arts) for the biological illustration in Fig. 1a. This work was funded by the Intramural Research Program of the NIH, National Cancer Institute, Center for Cancer Research.

## Author contributions

O.C. and O.L. conceived the project, designed the experiments, developed the model, analyzed the data, and wrote the manuscript. O.C. implemented the code. O.L. summarized the results, figures, and tables. M.R.G. contributed to the experiment design and writing of the manuscript. All authors read and approved the manuscript.

## Competing interests

The authors declare no competing interests.
