## [Peer Review File · Nature Communications]

Reviewers' comments:

Reviewer #1 (Remarks to the Author):

This paper addresses the question of how Glioblastoma cells migrate and adapt to new environments during the process of tumour growth and progression. This is a question of highest clinical importance, and the authors study it using an unconventional and novel approach. Using newly available public data from various tumour anatomical regions, they apply a recently developed concept of exploratory adaptation and ask whether such a process could explain the dynamic trajectory of the cancer. Putting this theoretical concept to practical test in the context of cancer is important: if relevant it may shed new light of how the basic understanding of the disease. In the case of GBM, its properties – high heterogeneity, adaptability to new environments and development of drug resistance – suggest that they are particularly suitable candidates for such a process.

Extending the model of exploratory adaptation, the authors define phenotypes and phenotypic distances, and compute from the data the phenotypic trajectories followed by the cancer cells. Using these tools they identify trajectories, study their reversibility and whether exploratory adaptation is required to follow these trajectories. Finally they suggest connections between their finding and therapeutic strategies.

One fundamental problem with the design of the study is the following: exploratory adaptation should in principle allow finding various solutions to unforeseen challenges. In their model the authors test whether a specific distribution of solutions found can be reached by random variation of genetic interactions. However, it is possible that in their simulation a different solution was reached that does not match the one observed. This problem is related to the question of sample identity (see below).

The novelty of the concept and its non-traditional applicability to cancer data by itself is interesting enough to merit publication in a journal of general readership and high impact. However, the paper is very difficult to read. The details are described using too much bioinformatic jargon and not explained clearly enough. The analysis contains many delicate definitions (phenotypes, distances etc) that are not presented in a self-contained way or discussed, neither are many modelling and analysis choices justified. Understanding the results depends crucially on these details.

In principle, it should be possible to reproduce the analysis from the publicly available data and the manuscript. Below are specific comments and requirements for clarification that would enable this.

1. Gene list and connections : the network used in this study (~4000 genes and ~78000 directed connections) need to be specified, possibly in supplementary data.

2. Ivy-GAP data: the Methods section states 270 samples and then 122 samples (divided into locations that actually add up to 123). Which is the right number? How many samples were used in the study? Are the 42 tumours pooled from different patients? if so, how many?

3. Pathway activity: this quantity needs to be defined – it is not enough to quote an R library. What exactly is being computed?

4. Pathway distribution distance: this is computed as a Euclidean distance between two distributions (Minkowski here is misleading). Here the question of the sample identity becomes important: do the samples contain pooling of different patients? In that case, why does it make sense to first compute a distribution and then the distance, rather than computing the distance between samples of the same patient first, and then doing the statistics?

Moreover: Why are the distances computed per-sample in the simulation but not in the data?

On p.11 , Methods, there is reference to GD, is this meant to be GDD defined earlier?

5. Location specific phenotypes: are different patients pooled?

6. Fig. S1 does not have a caption. What is plotted there and how exactly was it computed? What is $d=0.5$?

7. All supplementary figures need detailed captions. Clustering algorithms and parameters need to be specified, etc.

Regarding the model of exploratory adaptation, why is it necessary to define the convergence criterion for each pathway separately? What happens if only the Global distance measure is considered?

Simulation of familiar environment: What are the initial conditions to the simulation? Is it the measured phenotype? if so, then this is really a consistency test of the dynamic model. Edge strength is taken from correlations between genes, and this provides input to the model; then, you are really asking whether all of this is consistent and the cells evolve to stay in the vicinity of their measured phenotype. If this is not so, please explain.

"... three main trajectories are preserved under familiar conditions (Fig. S1)" – how is this seen in Fig. S1?

The analysis of reverse trajectories is interesting, and could imply that while learning a novel phenotype, still the previous attractor is maintained. This point could be further discussed.

Reviewer #2 (Remarks to the Author):

1. What are the major claims of the paper?

The authors provide an interesting approach, which is very well implemented over an interesting use case, that may even provide clinical utility.

The authors use pathway metrics and exploratory adaptation to look into the possible differences between different GBM regions. Identifying such differences may provide, according to the authors, a way to intervene in a manner that prevent the transition between tumor states.

2. Are they novel and will they be of interest to others in the community and the wider field?

Yes, these claims are novel as far as I know. I have not seen a similar implementation of the approach.

3. Is the work convincing, and if not, what further evidence would be required to strengthen the conclusions?

It might be beneficial to examine some of the core assumptions made throughout the paper. For example, the work starts with a set of differentially expressed genes. There are multiple ways to obtain such a list, and it would be interesting to see if the work is robust in the context of other (perhaps one should be enough) methods.

Also, I assume that such a list of DE genes has been obtained in other works. Is the list in the current work similar to other lists?

The same goes for the tool used to obtain pathway activity. The authors used ssGSEA, but there are many ways to obtain pathway scores. If another method would have been used that would

lead to similar conclusions, it would greatly support the obtain biological understanding.

some more points;

I do not understand how this "The static snapshot of the system shows that smaller changes in phenotype are required..." Could be understood from the figure.

This may connect to some issues with Figure 1. For example, Figure 1D contains multiple sub-panels, which are hard to see. At first glance, it seems impossible to even know that there is text in the 1D panels.

The videos are a nice addition, but is it possible to use a format more friendly than GIF?

In principle, it seems as if it would be possible to see the enrichment of the different signatures of the pathway based stratification, of locations obtained by the authors, within TCGA tumors. It might be interesting to do so and to see if there is any association between such signatures and GBM phenotypes (even with the minimal "phenotype" of patient survival).

It might also be interesting to see any connection that might be obtained between the pathway signatures and immune signatures such as immunoscore.

Reviewer #1 comments to authors:

Reviewer #1: *This paper addresses the question of how Glioblastoma cells migrate and adapt to new environments during the process of tumour growth and progression. This is a question of highest clinical importance, and the authors study is using an unconventional and novel approach. Using newly available public data from various tumour anatomical regions, they apply a recently developed concept of exploratory adaptation and ask whether such a process could explain the dynamic trajectory of the cancer. Putting this theoretical concept to practical test in the context of cancer is important: if relevant it may shed new light on how the basic understanding of the disease. In the case of GBM, its properties – high heterogeneity, adaptability to new environments and development of drug resistance – suggest that they are particularly suitable candidates for such a process.*

Extending the model of exploratory adaptation, the authors define phenotypes and phenotypic distances, and compute from the data the phenotypic trajectories followed by the cancer cells. Using these tools they identify trajectories, study their reversibility and whether exploratory adaptation is required to follow these trajectories. Finally, they suggest connections between their findings and therapeutic strategies.

One fundamental problem with the design of the study is the following: exploratory adaptation should in principle allow finding various solutions to unforeseen challenges. In their model the authors test whether a specific distribution of solutions found can be reached by random variation of genetic interactions. However, it is possible that in their simulation a different solution was reached that does not match the one observed. This problem is related to the question of sample identity (see below).

The novelty of the concept and its non-traditional applicability to cancer data by itself is interesting enough to merit publication in a journal of general readership and high impact. However, the paper is very difficult to read. The details are described using too much bioinformatic jargon and not explained clearly enough. The analysis contains many delicate definitions (phenotypes, distances etc) that are not presented in a self-contained way or discussed, neither are many modelling and analysis choices justified. Understanding the results depends crucially on these details.

Reply: A large gap exists between theoretical modeling and the use of such models with patient data to obtain clinically-relevant insights. This is partly due to the multidisciplinary nature of computational biology research, which necessitates the use of applied mathematics, theoretical physics, bioinformatics, and data science, and the marrying of these different mindsets and diverse ways of framing a problem. We are interested in advancing the cancer modeling field and being able to explore important questions using the combined strength of theoretical and data-driven approaches. Our goal is for the manuscript to

reach audiences from all these communities, with the hope that they could all further use it. We have attempted to clarify and simplify the manuscript where possible.

In this study we tested a theoretical modeling framework with complex real-patient data that include spatial tumor information. Since obtaining large scale time series molecular profiles of patient tumors is next to impossible in the GBM context, we had to propose a creative dynamical model that could simulate such temporal behavior. The main question that we have addressed is whether the observed transitions between location-based phenotypes could be explained, in part, by the EA process. We used the data from one location as the initial condition and simulate the time evolution in phenotype; we also assess the proximity of such derived phenotypes from other observed location phenotypes. As we elaborate throughout the paper, there are cases (transition trajectories) in which the EA process is helpful to some degree, and others in which it is not. From our results, one can infer that other steady states (phenotypes) may be possible in proximity of the Ivy-GAP patient phenotypes. If data from finer spatial sampling can be obtained in the future, it may be possible to observe such new predicted phenotypes. Since, at the moment, we do not have such supporting data, we refrain from making any further conclusions in this regard.

We provide below details on how the data, the implementation, and guiding examples will be made available. This should make it possible to reproduce the analysis.

Reviewer #1: *1. Gene list and connections: the network used in this study (~4000 genes and ~78000 directed connections) need to be specified, possibly in supplementary data.*

Reply: The data (processed gene expression, phenotype labels, gene lists, pathway ontology, computed pathway activity, and TF-gene regulatory networks) are now included in an open access Github repository that will be shared once the manuscript is accepted, together with the source code of the implementation and guiding examples.

Reviewer #1: *2. data: the Methods section states 270 samples and then 122 samples (divided into locations that actually add up to 123). Which is the right number? How many samples were used in the study? Are the 42 tumours pooled from different patients? if so, how many?*

Reply: We would like to thank the reviewer for finding this mistake. We mistakenly listed in the manuscript CTpnz (26) instead of CTmvp (25), leading to the discrepancy in numbers noted by the reviewer.

The Ivy-GAP is the most comprehensive study of anatomic structures of glioblastomas to date. The project profiled 42 tumors donated from 41 patients. *In situ* hybridization (ISH) with genes selected according to their known or presumed relevance to GBM was used to screen for gene expression enriched in particular structures and cell clusters, and laser microdissection followed by RNA sequencing were used to generate the transcriptomes and identify genetic markers. The anatomic structures part of the study included a primary survey of 8 tumors, a comprehensive survey of 10 tumors with the largest number of structures in the least number of blocks that generated 122 samples split by anatomic structure as follows: CT (30),

CTmvp (25), CTpan (24), IT (24), LE (19), and a further ISH screen of genes identified by RNASeq in 29 tumors.

The samples per tumor and their locations are summarized in the following table, which is now also included in the supplemental tables (Table S1); the manuscript is also clarifying some of this information:

Tumor ID	GBM Subtype	MGMT Meth	Survival Days	EGFR Amp	KPS	Age	CT	CTmvp	CTpan	IT	LE
W11-1-1	Classical, Mesenchymal	Yes	1076	No	100	57	3	3	3	2	2
W1-1-2	Classical	No	105	Yes	100	66	3	3	3	3	3
W2-1-1	Classical, Neural	Yes	1096	Yes	90	64	3	3	3	3	2
W26-1-1	Neural	Yes	1293		100	57	3	0	3	0	0
W31-1-1	Proneural	No	871	No	90	17	3	3	0	3	3
W32-1-1	Proneural	Yes	NA	No	90	56	3	3	3	0	0
W5-1-1	Classical, Neural	No	NA	Yes	90	64	3	3	3	3	2
W55-1-1	Classical	Yes	NA		100	52	3	3	0	3	3
W8-1-1	Classical, Mesenchymal	No	442	No	70	49	3	1	3	3	3
W9-1-1	Proneural	No	145	No	90	50	3	3	3	4	1

Reviewer #1: 3. Pathway activity: this quantity needs to be defined – it is not enough to quote an R library. What exactly is being computed?

Reply: Throughout the study we attempt to use standard, well-established bioinformatics approaches in an effort to increase the reproducibility of the results and the generality of our framework. Gene set enrichment (GSE) analysis is one such framework: it enables condensing information from gene expression level to functional pathways or signatures summary, thereby reducing the noise and dimensions of the data. Such functional pathway summary, also known as pathway activity or pathway enrichment score, captures the degree to which the genes involved in a pathway are coordinately up or down regulated. Many GSE methods are supervised and population based, in that they compute enrichment scores as comparisons between two groups of the population (for example, case/control). However, our simulation approach, tracking changes at a single sample level, necessitates the use of an enrichment approach that can be applied at the single sample level, and without regard to the class labeling. ssGSEA is a non-parametric, unsupervised method, which starts by evaluating the rank of each gene per sample; for each pathway/gene set a Kolmogorov-Smirnov-like rank statistic is calculated, giving an estimate of the overall rank of the genes belonging to that pathway compared to the rank of the rest of the genes ¹. Hänzelmann et al. compare several competing GSE methods, including ssGSEA and their

own method GSVA, and find that these two outperform the others both with respect to sensitivity and accuracy of the methods to identify differential pathway activity². We have elaborated this further in the manuscript.

Reviewer #1: *4. Pathway distribution distance: this is computed as an Euclidean distance between two distributions (Minkowski here is misleading). Here the question of the sample identity becomes important: do the samples contain pooling of different patients? In that case, why does it make sense to first compute a distribution and then the distance, rather than computing the distance between samples of the same patient first, and then doing the statistics?*

Moreover: Why are the distances computed per-sample in the simulation but not in the data?

On p.11, Methods, there is a reference to GD, is this meant to be GDD defined earlier?

Reply: The samples per location are indeed pooled from different patients. There are several reasons for following this approach. First, the Ivy-GAP dataset contains multiple samples from the same tumor for some of the regions, while no samples for others; so multiple possibilities and sometimes none exist for mapping these samples into initial-final pairs per patient. Second, GBMs have been shown to exhibit intra-tumoral heterogeneity at the single cell resolution; Patel et al., for example, showed through single cell transcriptomic profiling that primary GBMs display a mixture of the four established GBM transcriptomic subtypes³, with individual cells from the same tumor displaying one of the four subtypes⁴. Indeed, as seen from the molecular characteristics of the tumors presented in the summary table above, some of the tumors display a mixture of transcriptomic subtypes (Classical + Mesenchymal, or Classical + Neural, for example), confounding the question of which of the samples ‘moved’ to a different location. Last, the location-centric distributions capture the location specific phenotypes, which as shown both in the Ivy-GAP landmark paper as well as our supplementary figures (pathway heatmaps) are quite distinct. Therefore, it is reasonable to assume that there exist such distributions at the location level across patients.

An informative description of a phenotype can be made by a high-dimensional vector of gene expressions; as described, a lower-dimension pathway activities vector can also be used to characterize the phenotype. However, as explained, a data-derived location phenotype is, in fact, characterized as a vector of pathway distributions. To compute the distance between two such phenotypes we compute for each pathway the histogram distance between the two distributions, which we call PDD; as the reviewer points out, this is indeed the Euclidean distance between the distributions -- the reference to “Minkowski distance” refers to the specific implementation in R, but has now been removed from the main text and mentioned only in the Methods section. A global distance of the two phenotypes is computed as the sum of PDDs over all the pathways, which we call GDD. Our goal in this first data-driven part of the study was to study the actual GBM patient data in a manner relevant to our scientific question, and without any modeling assumptions.

In the second, theory-driven part we develop the modeling framework that will enable reasoning about the behavior of single cells; the goal of this part is to assess whether EA-mimicking behavior of cells can explain the observable differences characterized in part 1. As we are now operating at the single cell level,

we propose the notion of PD, which captures how far a single cell's pathway activity is from that pathway's distribution for a given location. A global measure of a single, simulated cell's phenotypic distance from a location phenotype is given by summing the PDs over all pathways; this is what we call GD. Being able to compute sample-specific distances from a target phenotype becomes useful when modeling a gradient-descent-like exploration.

We have attempted to clarify this further in the manuscript, as well as to fix erroneous references to GD or GDD.

Reviewer #1: *5. Location specific phenotypes: are different patients pooled?*

Reply: Yes, please see our reply to Comment #4.

Reviewer #1: *6. Fig. S1 does not have a caption. What is plotted there and how exactly was it computed? What is $d=0.5$? All supplementary figures need detailed captions. Clustering algorithms and parameters need to be specified, etc.*

Reply: Captions to all figures, and all relevant information, are now included in the manuscript and supplementary files.

Reviewer #1: *8. Regarding the model of exploratory adaptation, why is it necessary to define the convergence criterion for each pathway separately? What happens if only the Global distance measure is considered?*

Reply: Microenvironment stimuli are well characterized to be non-uniformly distributed and gradient-based^{5,6}. In particular, in a 3D tumor mass, a gradient of oxygen or hypoxia is established by the combined effects of cellular metabolism and oxygen diffusion; this gradient-based nature of the microenvironment stimuli, is in fact, a major barrier to recapitulating the *in vivo* conditions in *in vitro* models⁷⁻⁹. The Blood Brain Barrier/Blood Tumor Barrier introduce further osmotic gradients. Thus, uniform response may not be the most biologically-relevant assumption in the GBM case. Since, different stressors (nutrient depletion/hypoxia/osmotic pressure) may perturb different mechanisms, and thus activate different pathways, the global compression value may not be sufficient to describe the phenotypic changes. We therefore coupled global measurements with more detailed pathway activity vectors.

Moreover, uniform exploratory ability did not lead to convergence of phenotypes even in cases where we would expect a degree of convergence for some simulated cases (for example, along CT->IT->LE). We therefore introduced the ability to explore a high dimensional search space in a biased, or gradient-like, manner, which we expect to have biological and functional relevance. We implemented an exploratory process that starts with an initial stochastic white noise scaled with a parameter that changes over time as a function of the pathway vector distance to its presumed target. Our pathway-centric convergence this way enables convergence along pathways that are more essential for the main stressor in question and gradual coverage of other pathways less essential to adaptation to that spatial location. We were

interested in studying pathways that managed to converge, and ultimately their impact on the global measure of the overall system.

Reviewer #1: *9. Simulation of familiar environment: What are the initial conditions to the simulation? Is it the measured phenotype? if so, then this is really a consistency test of the dynamic model. Edge strength is taken from correlations between genes, and this provides input to the model; then, you are really asking whether all of this is consistent and the cells evolve to stay in the vicinity of their measured phenotype. If this is not so, please explain.*

Reply: Yes, the reviewer is correct. The initial conditions are the patient data on a specific tumor location, and the data on the target location were compared with our simulation results. All results are based on real patient data integrated with dynamical modeling. Such approach of integration is still missing in most cancer research and is one of the main contributions of our effort.

Reviewer #1: *"... three main trajectories are preserved under familiar conditions (Fig. S1)" – how is this seen in Fig. S1?*

Reply: From BGA and heatmap cluster plotted in Figure S1, one can see that the CT phenotype is the center of all locations. From CT, there are 3 closest locations, CTmvp, CTpan, and IT; these are separated from each other. From IT, LE is the next closest location. Thus, we have three immediate trajectories, CT->IT->LE, CT->CTmvp, CT->CTpan.

Reviewer #1: *10. The analysis of reverse trajectories is interesting, and could imply that while learning a novel phenotype, still the previous attractor is maintained. This point could be further discussed.*

Reply: We thank the reviewer for this interesting suggestion. It is now included in the discussion section.

Reviewer #2 (Remarks to the Author):

1. What are the major claims of the paper?

Reviewer #2: *The authors provide an interesting approach, which is very well implemented over an interesting use case, that may even provide clinical utility. The authors use pathway metrics and exploratory adaptation to look into the possible differences between different GBM regions. Identifying such differences may provide, according to the authors, a way to intervene in a manner that prevents the transition between tumor states.*

2. Are they novel and will they be of interest to others in the community and the wider field?

Reviewer #2: *Yes, these claims are novel as far as I know. I have not seen a similar implementation of the approach.*

3. Is the work convincing, and if not, what further evidence would be required to strengthen the conclusions?

Reviewer #2: *It might be beneficial to examine some of the core assumptions made throughout the paper. For example, the work starts with a set of differentially expressed genes. There are multiple ways to obtain such a list, and it would be interesting to see if the work is robust in the context of other (perhaps one should be enough) methods.*

Also, I assume that such a list of DE genes has been obtained in other works. Is the list in the current work similar to other lists?

Reply: To ensure that our results are robust and reproducible and to increase the generality and use of our framework we attempt throughout the study to use standard and well-established bioinformatics approaches. To calculate the DE gene list, we used the recommended TCGA workflow ¹⁰ including the built-in functions that TCGAAbiolinks ¹¹ provides for preprocessing, normalization, and differential expression analysis; in particular, these take into account batch effects and other TCGA data specifics. Moreover, as our primary reason for starting from a select set of genes was to reduce the dimensionality of the data, we further expanded the list of genes, to include genes whose expression was variable in the Ivy-GAP set, as well as any transcription factors, regardless of their expression levels. Such a step creates a list that includes approximately 1/3 of the genes that are sufficiently expressed in the Ivy-GAP cohort.

We assessed the overlap of our gene list with other glioma related sets by selecting the (25) glioma related gene sets curated as part of the Molecular Signatures Database (MSigDB, <http://software.broadinstitute.org/gsea/msigdb>) ¹². For each of these MSigDB gene sets we show the percentage of expressed genes that overlap with our selected gene list. We are encouraged to see that the majority of the glioma relevant gene sets are well represented in our selected list (is now included in Table S1).

MSigDB Geneset	%Present
COLIN_PILOCYTIC_ASTROCYTOMA_VS_GLIOBLASTOMA_DN	70.4
YAMANAKA_GLIOBLASTOMA_SURVIVAL_UP	66.7
VERHAAK_GLIOBLASTOMA_MESENCHYMAL	61.6
MUELLER_METHYLATED_IN_GLIOBLASTOMA	57.1
ZHENG_GLIOBLASTOMA_PLASTICITY_DN	53.6
VERHAAK_GLIOBLASTOMA_PRONEURAL	53
JOHANSSON_GLIOMAGENESIS_BY_PDGF_UP	51.8
ZHENG_GLIOBLASTOMA_PLASTICITY_UP	50.4
COLIN_PILOCYTIC_ASTROCYTOMA_VS_GLIOBLASTOMA_UP	48.6
VERHAAK_GLIOBLASTOMA_NEURAL	41.7
NUTT_GBM_VS_AO_GLIOMA_UP	41.3
BEIER_GLIOMA_STEM_CELL_UP	41.2

BEIER_GLIOMA_STEM_CELL_DN	41
TCGA_GLIOMASTOMA_COPY_NUMBER_UP	40.9
KEGG_GLIOMA	40
JOHANSSON_GLIOMAGENESIS_BY_PDGFBN_DN	40
ROVERSI_GLIOMA_COPY_NUMBER_UP	39.8
NGO_MALIGNANT_GLIOMA_1P_LOH	37.5
TCGA_GLIOMASTOMA_MUTATED	37.5
NUTT_GBM_VS_AO_GLIOMA_DN	37.2
ROVERSI_GLIOMA_COPY_NUMBER_DN	36.7
VERHAAK_GLIOMASTOMA_CLASSICAL	36.5
ROVERSI_GLIOMA_LOH_REGIONS	34.3
TCGA_GLIOMASTOMA_COPY_NUMBER_DN	29.6
YAMANAKA_GLIOMASTOMA_SURVIVAL_DN	25

Reviewer #2: *The same goes for the tool used to obtain pathway activity. the authors used ssGSEA, but there are many ways to obtain pathway scores. If other methods would have been used that would lead to similar conclusions, it would greatly support the obtain biological understanding.*

Reply: Please see our detailed reply to reviewer 1' question #3.

Reviewer #2: *some more points;*

I do not understand how this "The static snapshot of the system shows that smaller changes in phenotype are required..." Could be understood from the figure.

This may connect to some issues with Figure 1. For example, Figure 1D contains multiple sub-panels, which are hard to see. At first glance, it seems impossible to even know that there is text in the 1D panels.

Reply: Figure 1 introduces all concepts, types of data that we worked with throughout the paper, and the calculated phenotype distances based on data (not modeling). This is a static view of the tumor system, as no information regarding time is included in the Ivy-GAP dataset. The first two subplots in Figure 1D are enlarged in Figure S1. Subplot D1 is the heatmap cluster of locations, subplot D2 is the BGA (similar to PCA), and subplot D3 is the calculated global phenotype distance between every pair of locations. From these 3 subplots, one can see that CT is closest to CTpan and to IT. Next in distance, CT is closer to CTmvp, than to LE. We note that if one would calculate the distance from CT to LE, they would realize that this distance is large. However, the gaps in phenotype between CT to IT, and from IT to LE are small; indeed the spatial proximity of such pairs suggests that fewer changes would be needed to bridge these gaps.

Reviewer #2: *The videos are a nice addition, but is it possible to use a format more friendly than GIF?*

Reply: We have now provided mp4 videos which we hope will provide a more user-friendly exploration of pathway activity changes over time.

Reviewer #2: *In principle, it seems as if it would be possible to see the enrichment of the different signatures of the pathway based stratification, of locations obtained by the authors, within TCGA tumors. It might be interesting to do so and to see if there is any association between such signatures and GBM phenotypes (even with the minimal "phenotype" of patient survival).*

Reply: There are many interesting questions and further investigations that would be important to ask following our fundamental work. The reviewer gives two important examples of such extensions to our work: implications of spatial EA on patient survival and relationship to immune response. For now our focus has been on developing the framework, and establishing the study of the dynamics of phenotype transition pertaining to spatial heterogeneity. However, it would certainly be interesting to look at whether the static snapshot and enrichment for different location signatures can inform on characteristics such as tumor aggressivity. Preliminarily, the clinical information and GBM subtypes associated with the IVY tumors (see the newly added supplemental table with clinical and molecular information) suggest that the distinct anatomical structures can be observed in tumors of all GBM subtypes, although interestingly the patient with the longest survival harbored a tumor in which no LE, IT, or CTmpv could be sampled. Whether this is primarily due to such patients harboring smaller tumors or rather a consequence of the molecular characteristics of the present anatomic locations requires further analysis that is outside the scope of this paper.

Reviewer #2: *It might also be interesting to see any connection that might be obtained between the pathway signatures and immune signatures such as immunoscore.*

Reply: We would like to thank the reviewer for the interesting suggestion. Indeed, immune response is a paramount subject in GBM research. GBMs are considered mostly 'immunologically cold' tumors: they arise in the immunologically-privileged Blood Brain Barrier (BBB)-protected environment of the brain, and secrete factors that inhibit both the innate and adaptive immune systems. However, due to the poorly-formed neo-vasculature of GBMs, the BBB and Blood Tumor Barrier (BTB) can often be leaky, so it is reasonable to expect some level of immune cell infiltration.

We used xCell¹³ to deconvolute the (whole genome) bulk expression profiles of the different Ivy-GAP locations, and obtained the following signatures.

The most interesting microenvironment appears to be around microvascular proliferation, which suggests that the BBB/BTB is indeed leaky (due to no tight junctions): we see endothelial signatures, but also fibroblasts, pericytes, and some enrichment of immune signatures. This suggests that there is trafficking of immune cells in the microenvironment, but the paucity of these cells in other regions of the tumor (as seen by the weak signal from other locations) suggests that few resident or trafficked immune cells are present in the other regions. The recently announced failure to meet the primary endpoint of a phase 3 clinical trial evaluating the programmed death (PD1) immune checkpoint inhibitor Nivolumab in combination with radiation for newly diagnosed GBM patients with unmethylated MGMT (CheckMate-498 (NCT02617589) ¹⁴) also indicates that additional therapeutic interventions that increase trafficking of immune cells are needed to complement any immune-checkpoint inhibitor approaches. We will include this analysis in the Results section.

References

1. Barbie, D. A. *et al.* Systematic RNA interference reveals that oncogenic KRAS-driven cancers require TBK1. *Nature* **462**, 108–112 (2009).
2. Hänzelmann, S., Castelo, R. & Guinney, J. GSVA: gene set variation analysis for microarray and RNA-seq data. *BMC Bioinformatics* **14**, 7 (2013).
3. Verhaak, R. G. W. *et al.* Integrated genomic analysis identifies clinically relevant subtypes of glioblastoma characterized by abnormalities in PDGFRA, IDH1, EGFR, and NF1. *Cancer Cell* **17**, 98–

- 110 (2010).
4. Patel, A. P. *et al.* Single-cell RNA-seq highlights intratumoral heterogeneity in primary glioblastoma. *Science* **344**, 1396–1401 (2014).
 5. Gerling, M. *et al.* Real-time assessment of tissue hypoxia in vivo with combined photoacoustics and high-frequency ultrasound. *Theranostics* **4**, 604–613 (2014).
 6. Gupta, S., Roy, A. & Dwarakanath, B. S. Metabolic Cooperation and Competition in the Tumor Microenvironment: Implications for Therapy. *Front. Oncol.* **7**, 68 (2017).
 7. Ando, Y. *et al.* A Microdevice Platform Recapitulating Hypoxic Tumor Microenvironments. *Sci. Rep.* **7**, 15233 (2017).
 8. Bauer, N., Liu, L., Aleksandrowicz, E. & Herr, I. Establishment of hypoxia induction in an in vivo animal replacement model for experimental evaluation of pancreatic cancer. *Oncol. Rep.* **32**, 153–158 (2014).
 9. Hubert, C. G. *et al.* A Three-Dimensional Organoid Culture System Derived from Human Glioblastomas Recapitulates the Hypoxic Gradients and Cancer Stem Cell Heterogeneity of Tumors Found In Vivo. *Cancer Res.* **76**, 2465–2477 (2016).
 10. Silva, T. C. *et al.* TCGA Workflow: Analyze cancer genomics and epigenomics data using Bioconductor packages. *F1000Res.* **5**, (2016).
 11. Colaprico, A. *et al.* TCGAbiolinks: an R/Bioconductor package for integrative analysis of TCGA data. *Nucleic Acids Res.* **44**, e71 (2016).
 12. Liberzon, A. *et al.* The Molecular Signatures Database (MSigDB) hallmark gene set collection. *Cell Syst* **1**, 417–425 (2015).
 13. Aran, D., Hu, Z. & Butte, A. J. xCell: digitally portraying the tissue cellular heterogeneity landscape. *Genome Biol.* **18**, 220 (2017).
 14. Bristol-Myers Squibb Announces Phase 3 CheckMate -498 Study Did Not Meet Primary Endpoint of

Overall Survival with Opdivo (nivolumab) Plus Radiation in Patients with Newly Diagnosed MGMT-Unmethylated Glioblastoma Multiforme | BMS Newsroom. Available at:
<https://news.bms.com/press-release/corporatefinancial-news/bristol-myers-squibb-announces-phase-3-checkmate-498-study-did>. (Accessed: 13th June 2019)

REVIEWERS' COMMENTS:

Reviewer #1 (Remarks to the Author):

Some of the Reviewers' comments were satisfactorily addressed. Specifically questions regarding the data and its relation to the model were clarified.

However, the general comment that the paper is difficult to read by a broad audience still remains (though to a lesser extent). I would urge the authors to make additional effort to clarify their results and presentation. Analyses should be explained clearly; figures should be described clearly, referenced from the text and tightly linked to it, etc.

Here are some specific remarks that might help towards this end.

- Fig. 1E: what do the solid/dashed lines represent?
- Fig. 3C: Here there are two types of edges (here differing by color) that are not explained. (one may guess that this is another way of marking the differences from Fig. 1, in addition to the + signs, but this is not clearly explained).
- "We also assessed whether exploratory adaption is necessary for this reversal by simulating process reversal without stochastic perturbations of the regulatory network." How exactly do you simulate reversal in this case? In Fig. 4A, are the results shown for the exploratory adaptation calculations? Should be clearly stated.
- Please refer to Fig. 4B in the main text and explain what is presented in it and how it supports the claim of the text. In this figure, what are the circles that do not have arrows ? what do circles with arrows of the same color represent? it is also not clear what is the difference between the three panels.
- The paragraph on "Discretizing the continuous phenotype" remains very unclear. The idea of a continuum of states in between two stable phenotypes is appealing (and appears in other biological contexts). However, the actual computation is described in a sketchy way and the conclusion not developed: the indication is that a continuum of phenotypes is a natural part of the transition. Table S2 was not found in the supplementary file, and maybe if it only contains a handful of numbers (distances following deletions of samples) these can be quoted in the main text.

The new analysis on immune signatures of the phenotype looks like an interesting lead to another project – it is not clear how it relates to the concept developed in the current work.

Having said that, I am still with the opinion that this work is interesting and highly novel and therefore worthy of publication.

Reviewer #2 (Remarks to the Author):

The authors answered most of my questions and made some of the changes I suggested. I respect the choices they made in not making some of the changes and do not ask for any other changes.

REVIEWERS' COMMENTS:

Reviewer #1 (Remarks to the Author):

Some of the Reviewers' comments were satisfactorily addressed. Specifically, questions regarding the data and its relation to the model were clarified.

However, the general comment that the paper is difficult to read by a broad audience still remains (though to a lesser extent). I would urge the authors to make additional effort to clarify their results and presentation. Analyses should be explained clearly; figures should be described clearly, referenced from the text and tightly linked to it, etc.

Here are some specific remarks that might help towards this end.

- Fig. 1E: what do the solid/dashed lines represent?

Reply: Caption of Fig. 1 is now adjusted.

- Fig. 3C: Here there are two types of edges (here differing by color) that are not explained. (one may guess that this is another way of marking the differences from Fig. 1, in addition to the + signs, but this is not clearly explained).

Reply: Caption of Fig. 3 is now adjusted.

- "We also assessed whether exploratory adaption is necessary for this reversal by simulating process reversal without stochastic perturbations of the regulatory network." How exactly do you simulate reversal in this case?

Reply: This sentence refers to simulation of reversal without perturbing the weights of the TF-gene regulatory network. This would capture an acquired, intrinsic ability to revert back to the original

phenotype.

In Fig. 4A, are the results shown for the exploratory adaptation calculations? Should be clearly stated.

Reply: We clarify these points as follows: the caption notes “The reverse process of transitioning between phenotypes was simulated for each edge of the location-based phenotype network from Figure 3”. In addition, in the relevant Results section we clarify that these are simulation results: “To address this question, for each original exploration (for example, starting from CT with target CTmvp, with resulting GD (CT→CTmvp)=13.83) we simulated the reverse exploration (starting from CTmvp with target CT, with resulting GD (CT←CTmvp)=9.15) such reverse processes, by setting the initial condition to the end location of the original direction; we then compared the resulting distances, for example, the final distance resulting from exploration LE→CT to the final distance resulting from exploration CT→LE (see Figure 4A, Table S2 Supplementary Table 2).” To emphasize that this is a simulation-based result, we have removed the word ‘calculated’ and now use ‘simulated’.

- Please refer to Fig. 4B in the main text and explain what is presented in it and how it supports the claim of the text. In this figure, what are the circles that do not have arrows? what do circles with arrows of the same color represent? it is also not clear what is the difference between the three panels.

Reply: Caption of Fig. 4 and related text are now adjusted.

- The paragraph on “Discretizing the continuous phenotype” remains very unclear. The idea of a continuum of states in between two stable phenotypes is appealing (and appears in other biological contexts). However, the actual computation is described in a sketchy way and the conclusion not developed: the indication is that a continuum of phenotypes is a natural part of the transition. Table S2 was not found in the supplementary file, and maybe if it only contains a handful of numbers (distances following deletions of samples) these can be quoted in the main text.

Reply: The table is now included. We considered whether removing some of the intermediate states of the continuum (removing several samples from the origin and target phenotypes) would have an effect on the final phenotypes reached by simulation. We observed that removing the most intermediate phenotypes (2-3 samples from the origin and target that were closest to the other phenotype) resulted in a final distance that was larger as compared to when the same number of samples were removed.

The new analysis on immune signatures of the phenotype looks like an interesting lead to another project – it is not clear how it relates to the concept developed in the current work.

Reply: We have clarified the relevance of this section to the project in the manuscript. Briefly, we considered the question of adaptability with phenotypes defined in terms of immune signatures, rather than functional pathway activities. However, as seen by the results, with few noted exceptions, most of the locations display immunologically-cold profiles.

Having said that, I am still with the opinion that this work is interesting and highly novel and therefore worthy of publication.

Reviewer #2 (Remarks to the Author):

The authors answered most of my questions and made some of the changes I suggested. I respect the choices they made in not making some of the changes and do not ask for any other changes.